# Silica Gel Impregnated by Deep Eutectic Solvents for Adsorptive Removal of BTEX from Gas Streams

**DOI:** 10.3390/ma13081894

**Published:** 2020-04-17

**Authors:** Patrycja Makoś, Edyta Słupek, Aleksandra Małachowska

**Affiliations:** Department of Process Engineering and Chemical Technology, Faculty of Chemistry, Gdansk University of Technology, G. Narutowicza St. 11/12, 80–233 Gdansk, Poland; edyta.slupek@pg.edu.pl (E.S.); aleksandra.malachowska@pg.edu.pl (A.M.)

**Keywords:** deep eutectic solvents, silica gel, adsorption

## Abstract

The paper presents the preparation of new adsorbents based on silica gel (SiO_2_) impregnated with deep eutectic solvents (DESs) to increase benzene, toluene, ethylbenzene, and p-xylene (BTEX) adsorption efficiency from gas streams. The DESs were synthesized by means of choline chloride, tetrapropylammonium bromide, levulinic acid, lactic acid, and phenol. The physico-chemical properties of new sorbent materials, including surface morphology and structures, as well as porosity, were studied by means of thermogravimetric analysis, Fourier transform infrared spectroscopy, scanning electron microscopy, X-ray diffraction, and Brunauer–Emmett–Teller analysis. The effect of DESs type, flow rate, and initial concentration of BTEX were also investigated followed by regeneration and reusability of adsorbents. The results indicate that SiO_2_ impregnated with tetrapropylammonium bromide and lactic acid in a 1:2 molar ratio have great potential for the removal of BTEX from gas streams. Its adsorption capacity was higher than the pure SiO_2_ and other developed SiO_2_-DES adsorbents. This result can be explained by the specific interaction between DESs and BTEX, i.e., hydrogen bonds interaction.

## 1. Introduction

Benzene homologs are widely used as raw materials, intermediates and organic solvents in many industrial processes; among which are benzene, toluene, ethylbenzene, and xylene (BTEX). They are commonly found in many waste streams, i.e., industrial wastewater [1] or waste gases [2]. They also occur on a large scale in natural gas [3] and biogas [4], from which they have to be removed before further processing due to their toxic nature and carcinogen character—shortening the lifetime of the catalysts, and having potential emission to the atmosphere during combustion processes [5,6].

The main technologies used for removal of BTEX from gas streams include absorption, conventional adsorption, pressure swing adsorption, membrane separation, cryogenic separation, and biological separation [2]. Chemical and physical absorption methods have several disadvantages i.e., foaming possibility, inefficiency for numerous groups of chemical compounds due to their limited solubility, and the high amount of chemical solvents (sometimes toxic and corrosive compounds) that they require. Pressure swing adsorption and cryogenic separation ensures the high efficiency of removing contaminants from gas streams. However, high electricity consumption, expensive investment and operation often prevent practical use. In addition, pre-treatment is required. Membrane separation is an expensive process and not suitable for high polluted gasses. In turn, biological separation is characterized by low impurities removal efficiency and longtime operation process. Among the available technologies, one of the most popular is adsorption, due to its high efficiency without use of toxic chemicals, with the possibility of adsorbents regeneration. In the industrial process, silica gel [5], activated carbon [7], carbon molecular sieves, zeolites [8], activated carbon impregnated with polymers, and polymeric adsorbents are the main adsorbents used for the removal of BTEX from the industrial gas streams [2]. Ideal adsorbents should be characterized by high selectivity and sorption capacity for BTEX as well as high stability after many adsorption-desorption cycles. However, as observed in several works, the sorption capacity and selectivity of popular adsorbents are often insufficient. One of the possible solutions is coating solid sorbents with high sorption capacity compounds i.e., 2,2′-(pentane-1,5-diylbis(oxy))dibenzaldehyde [9], monolithic polydimethylsiloxane [10], or 2,2′-(hexane-1,6-diylbis(oxy)) dibenzaldehyde [11]. In recent years, most of the work on the adsorbent surface modification has been focused on the use of ionic liquids (ILs) [12,13,14], due to their unique properties i.e., thermal and chemical stability, high adsorption capacity, low vapor pressure, non-flammable, non-volatile, and non-corrosive character, which can be changed by the selection of a cation and anion. Despite being ideal compounds for impregnating adsorbents, ILs have a few limitations, such as high price, tedious preparation process, toxic character, and problems with their recyclability and biodegradability.

Nowadays, ILs are replaced by a new type of green solvents named deep eutectic solvents (DESs). DESs are liquids compounds that are formed upon mixing two or more components that are involved in hydrogen bond or electrostatic interactions with each other to obtain a eutectic solution [15,16]. DESs are characterized by similar unique physicochemical properties to ILs, however their low price, low toxicity, and biodegradability make them potentially more beneficial compounds compared to ILs. Currently, DESs are widely used in analytical chemistry [17,18,19], desulfurization of diesel [20], lignocellulosic biomass treatment and detoxification [21,22], water [23], air [24] and biogas streams [25,26] treatment, and in catalysis [27]. In addition, DESs were successfully used for dearomatization of fuel using an extraction process. Based on the review of the scientific papers, it can be concluded that the most effective removal of BTEX from fuels is demonstrated by DES composed of quaternary ammonium salts and carboxylic acids or glycols, due to the possibility of COOH-π or OH-π bonds formation [28,29,30,31,32,33]. Although DESs have attracted large attention, the investigations concerning the impregnation of solid sorbents using DES for adsorptive purification of gas streams are limited. In several works the authors described the impregnation of activated carbon with the use of choline chloride:urea (1:2 molar ratio), silica gel with choline chloride:glycerol (1:2) [34], and choline chloride:urea (1:2 molar ratio) [35] and using them to remove carbon dioxide from the gas streams.

In this paper, new adsorbents were prepared using silica gel impregnated with DESs composed of choline chloride (ChCl), tetrapropylammonium bromide (TPABr) as hydrogen bond acceptors (HBAs), and levulinic acid (Lev), lactic acid (LA), phenol (Ph) as hydrogen bond donors (HBDs) to increase the BTEX adsorption efficiency from gas streams. Adsorbents based on SiO_2_ supported DES were characterized in terms of functional group content using Fourier transform infrared spectroscopy (FT-IR), stability and degradation by means of thermal gravimetric analysis (TGA), surface morphology using scanning electron microscopy (SEM), crystallinity using X-ray diffraction (XRD), and surface properties based on Brunauer–Emmett–Teller method (BET). The effect of several properties including the type of DES, flow rate, and initial concentration of BTEX on adsorption efficiency was studied, followed by an investigation of the regeneration and recycling of SiO_2_-DES. The BTEX adsorption capacity was compared with commercial silica gel.

## 2. Materials and Methods

### 2.1. Materials

For the synthesis of deep eutectic solvents and gas mixture preparations, reagents with high purity (purity ≥98%), i.e., choline chloride, tetrapropylammonium bromide, levulinic acid, lactic acid, phenol, benzene, ethylbenzene, toluene, and o-xylene, were purchased from Sigma-Aldrich (St. Louis, MO, USA). For the synthesis of adsorbents, commercial silica gel (particle size 7 µm) and methanol were purchased from POCH (Gliwice, Poland). Gases, i.e., nitrogen, hydrogen and air were with high purity N 5.0 or N 5.0 were purchased from Linde-Gas (Łódź, Poland).

### 2.2. Apparatus

In the studies, a gas chromatograph with flame ionization detector (GC-FID) Autosystem XL (PerkinElmer, Waltham, MA, USA) and capillary column HP-5 (30 m × 0.25 mm × 0.25 µm) was used to study BTEX adsorption processes. To study the structure of new adsorbents FT-IR-ATR Bruker Tensor 27 spectrometer (Bruker, Billerica, MA, USA) were used. The thermal analysis of samples was performed using the simultaneous thermal gravimetric (TG) and differential thermal gravimetric (TG/DTG) analyzer model TG 209 F3 Tarsus (company Netzsch, Selb, Germany). The morphological properties of the samples, topography were carried out using a scanning electron microscope FEI Quanta 250 FEG (Thermo Fisher Scientific, Waltham, MA, USA). The XRD analysis were done using the Rigaku Intelligent SmartLab X-ray diffraction device (Austin, TX, USA), equipped with a sealed x-ray generator, a copper shield operating at 30 mA and 40 kV. The Micromeritics Gemini instrument (model 2365) (Micromeritics, Norcross, GA, USA) was used for the analysis of surface area and total pore volume.

### 2.3. Methods

#### 2.3.1. Synthesis of Deep Eutectic Solvents

Deep eutectic solvents were synthesized by mixing ChCl or TPABr with Lev, LA, or Ph in 1:2 or 1:3 mole ratio at 70 °C until homogeneous liquids were obtained.

#### 2.3.2. Preparation of Silica Gel Modified by Deep Eutectic Solvents

Silica gel coated by deep eutectic solvents were prepared using incipient impregnation method at 25 °C, based on previous studies [36]. To prepare new adsorbents, silica gel was washed three times with methanol and dried in a vacuum oven at 110 °C for 12 h to remove impurities and water. Then, 6 g of DES was mixed with 1 g of methanol in a 50-mL vial followed by the addition of 10 g of silica gel. The adsorbents were agitated 2 h at 25 °C and dried at 90 °C for 6 h. Dried adsorbents were stored in a desiccator. The amount of DES loaded on the silica gel was determined using Equation (1):(1)$$m_{\mathrm{DES}}=m_{{\mathrm{SiO}}_{2}-\mathrm{DES}}-m_{{\mathrm{SiO}}_{2}}$$
where: m_SiO__2-DES_—mass of silica gel before DES impregnation (g); m_SiO__2_—mass of silica gel after impregnation by DES.

#### 2.3.3. Adsorbents Characterization

FT-IR-ATR spectra of new adsorbents and pure compounds were taken using following parameters: 4000–550 cm^−1^ of spectral range; 4 cm^−1^ of resolution; 256 of sample and background scans number and 0.5 cm of slit width.

For thermogravimetric analysis, 6 mg of samples were placed in a corundum dish. The study was conducted in N_2_ with a 100 mL/min flow rate in the temperature range of 35–700 °C with a temperature increase rate of 10 °C/min.

The morphological properties of the adsorbents were carried out using a SEM equipped with an ET detector (Everhart-Thornley Detector, Davis, CA, USA)—a secondary electron detector, that provided a high spatial resolution of about 1.2 nm at 30 kV. The apparatus allowed of obtaining high-quality images with the resolution of up to about 1 nm, which allowed the identification of materials and observation of the correlation between the components of the sample.

Samples of pure and modified silica gel were tested in the range of 5° to 80° in steps of 0.01° using XRD instrument. The sample scanning speed was 1°/min [37]. The crystal forms were determined in the vertical direction to the corresponding lattice plane based on the Scherrer’s equation [38]. The most intense peak achieved for each sample was used to determine the quantitative analysis using the RIR (Reference Intensity Ratio) method [37].

For the characterization of surface area and total pore, Brunauer–Emmet–Teller (BET) method was used, based on previous studies [37,38,39]. At first, the preparation procedure included weighing each sample in the amount of more than 0.1 g, pretreating by degassing with nitrogen at 200 °C for 2 h. Then, all of the samples were allowed to cool and weighed. The next step was testing at the liquid nitrogen temperature (77 K, −196.15 °C). For surface area determination, an adsorptive (N_2_) was dosed to the sorbent in controlled increments, the pressure was allowed to equilibrate and after each dose, and then the quantity adsorbed was calculated. Samples were tested, with 10 measurement points taken for each in about 1h 10 min. With the area covered by each adsorbed gas molecule and creating an average monolayer, the surface area was calculated. By extending the process and allowing the gas to condense in the pores, the sample’s pore volume was calculated.

#### 2.3.4. Adsorption and Regeneration Process

In the studies, the dynamic adsorption/desorption experimental set-up was used. Installation was equipped with a nitrogen bottle to create a model gas stream matrix. Nitrogen was split into two streams, one was directed to the column with BTEX to produce impurities gaseous by bubbling. The second stream was used to dilute polluted biogas to the expected concentration of BTEX. Then, the contaminated gas stream was directed to an adsorption column containing new adsorbents. The effectiveness of the adsorption process was controlled using gas chromatography. For this purpose, gas samples were taken before and after the adsorption column. BTEX concentration was measured by considering that adsorption equilibrium was attained when the BTEX concentration at the outlet of the adsorption column before and after adsorption process was almost equal to the BTEX concentration at the inlet. The breakthrough curves were expressed as the ratio (C_IN_/C_OUT_) of the initial concentration of BTEX (C_IN_) to the concentration of BTEX after the adsorption process (C_OUT_) according to the adsorption time (t).

The adsorption capacity (q) of new adsorbents was calculated using Equation (2):(2)$$q=\frac{F}{m}\cdot (C_{\mathrm{IN}}\cdot t-\int_{0}^{t}C_{\mathrm{OUT}}\mathrm{dt})\;[\mathrm{mg}/g]$$
where: F—total flowrate of gas mixture (m^3^/h); m—the adsorbent amount (g), t—time at which the adsorbent reaches saturation (h).

The desorption of BTEX was carried out by heating the adsorbent to 80 °C and introducing a nitrogen stream (5 L/h) to the column with the adsorbent.

#### 2.3.5. Chromatographic Analysis

The isotherm temperature oven in the gas chromatograph was 110 °C; injection port temperature was 250 °C, flow rate of carrier gas (N_2_) was 2 mL/min; injection mode was split 20:1; FID temperature was 250 °C; flow rates of detector gases were H_2_: 40 mL/min, air: 400 mL/min. The gas sample in volume of 0.5 mL was analyzed by gas chromatography.

## 3. Results and Discussion

### 3.1. Characterization of Adsorbents

The silica gel used in these studies is characterized by a high surface area and high thermal stability [40,41]. The high porosity of SiO_2_ supports the impregnation process due to the fact that the silica gel can bond with DES physically through the formation of hydrogen bonds or electrostatics interactions [42]. Studied DESs are characterized by high viscosity (>250 mPas), and a melting point below 25°C. The high viscosity of DESs is unfavorable in many applications, i.e., extraction, absorption. While in the process of impregnation, high viscosity enables permanent deposition of DES on the SiO_2_ surface. The list and characteristics of new adsorbents is presented in Table 1.

The experimental research on the mechanism of the impregnation of silica gel was done by FT-IR-ATR analysis. Obtained spectra of pure silica gel and pure DESs were compared with the spectra of impregnated silica gel (Figure 1a–d). All identified bands that can be attributed to phenol, levulinic acid, lactic acid, and quaternary ammonium salt are visible in the spectrum of the impregnated silica gel [45]. Characteristic bands corresponding to Si–O–Si symmetric and asymmetric vibration can be observed in the 1100 and 802 cm^−1^ wavenumbers in pure SiO_2_ and SiO_2_-DESs spectra. Additional peaks in the impregnated silica gel spectra can be observed in the range of 2973–2880, 2973–2917, 2973–2884, and 2976–2877 cm^−1^ (Figure 1a–d, respectively) that can be attributed to the stretching vibrations of C–H bonds from alkylammonium cations, phenol, and organic acids. The characteristic shifts of the carbon–carbon double bonds (C=C) towards the higher wavenumber can be observed in the Figure 1a at around 1635 cm^−1^, while the shifts of the carbonyl group (C=O) towards the higher wavenumber can be observed at around 1725, 1716, and 1735 cm^−1^ (Figure 1b,c,d, respectively), and the extra peaks in the ranges of 1605–1355, 1614–1371, 1720–1366, and 1628–1377 cm^−1^ (Figure 1a,b,c,d, respectively) were attributed to asymmetric and symmetric CH_2_ vibrations, C–O stretching bond, and C–H bonding vibration in the alkaline chain. Whereas, the peaks at 1605–1355 cm^-1^ were assigned to the C–O stretching bond and aromatic C–C stretching bond of the phenol ring, while peaks at 755–696 cm^−1^ can be assigned to aromatic C–H bending vibrations. This indicates that DESs have been successfully bonded on the surface of silica gel.

All adsorbents were submitted to thermogravimetric analysis and the obtained TG/DTG curves are presented in Figure 2. All new adsorbents showed thermal stablility in the range of 273–365 K. SiO_2_-TPABr:Lev had the highest thermal stability in the range of 273–296 K (Figure 2a), which indicates a large application potential because industrial adsorption/desorption processes are carried out in the range of 20 to 120 °C. Slightly lower thermal stability were observed for SiO_2_-ChCl: Lev and SiO_2_-TPABr: Lev (Figure 2a,b). In all TG curves, a slight weight loss can be observed between room temperature and 373.15 K. This is because of the evaporation of the residue physically absorbed methanol or/and water which was used for the impregnation process. After exceeding the upper value of thermal stability, sorbents began to degrade. In the first stage, the weight loss was caused by the loss of the HBD (Ph, Lev, LA) in DES structures. TG/DTG curves of impregnated SiO_2_ with use ChCl:Lev (1:2) showed the weight loss at the level of 13.82% in the range temperature between 380.65–518.53 K (Figure 2a), with use TPABr:Lev (1:3) the weight loss was at the level of 17.94% in the range temperature between 381.52–542.69 K (Figure 2b), with use TPABr:LA (1:2) showed the weight loss at the level of 12.20% in the range temperature between 395.85–542.69 K (Figure 2c) and with use TPABr:Ph (1:2) showed the weight loss at the level of 14.94% in the range temperature between 385.03–532.80 K (Figure 2d). The second degradation stage for SiO_2_-ChCl:Lev was observed in the temperature range between 518.53–735.05 K. The weight loss (3.55%) indicated the complete degradation of ChCl (Figure 2a). The weight loss of 1.99%, 3.18%, and 2.13% is observed in the temperature range 542.69–735.15, 542.69–735.15, and 532.80–735.15 K, respectively for SiO_2_-TPABr:Lev, SiO_2_-TPABr:LA, and SiO_2_-TPABr:Ph. The weight loss in the second step indicates the complete degradation of TPABr. Additionally, the peaks obtained on the DTG curve reflect the maximum reactive speed temperature associated with the change in adsorbent mass. The thermal decomposition temperatures of the modified silica gel are exothermic peaks. In all modified adsorbents was observed higher degradation temperature compared to pure DESs. The observed increases the thermal stability of which results from the bond between the silicon atom and the structure DES was formed.

The wide-angle XRD patterns of the SiO_2_ before and after modification by DESs are shown in Figure 3. The results indicate that the patterns of all the adsorbents have comparable broadband centered at around 22 which confirms the topological structure and amorphous nature of the silica gel [46,47,48]. In comparison to the silica gel before modification (black line), the intensity of peak in XRD patterns of all modified SiO_2_ is found to decrease along with line broadening. The decrease in intensity is probably caused by the filling of pores of the silica gel surface by DES structure. This filled a reduction in X-ray scattering contrast with maintaining the amorphous nature of silica is maintained even after modification.

SEM was used to explain the change in morphological features after the impregnation of silica gel with DESs. The SEM images captured at high magnification (Figure 4a) showed that the surface of the skeletal SiO_2_ was smooth before modification and after impregnation, the surface of SiO_2_ turned out to be rough (Figure 4b–e). In addition, on the surfaces of silica gel after all DESs impregnation, the agglomerations are witnessed. DESs agglomerations on the SiO_2_ surface are responsible for high sorption efficiency. The larger agglomeration area, provide to the greater the adsorption capacity [49]. The largest DES agglomerations were observed on the SiO_2_-TPABr-LA (Figure 4c) surface and slightly lower on the SiO_2_-TPABr-Lev (Figure 4d). Furthermore, the particle size of the impregnated phases was close to the backbone before impregnation process, which indicates that the silica gel has good mechanical strength.

The results of the BET surface area and total pore volume of SiO_2_ and SiO_2_ impregnated with DES are presented in Table 2. The results indicate that SiO_2_-DES have lower BET surface area, and pore volume in comparison to pure SiO_2_, which confirmed the impregnation occurred. This is because of the less N_2_ adsorbed by SiO_2_ after impregnation since the pores were filled with DESs making it have a lower amount of pores available for N_2_ adsorption thus lower the BET surface area. Similar results were also obtained in previous studies [34,49]. The smallest surface area and pore volume was observed for SiO_2_-ChCl:Lev (248.11 m^2^/g; 0.1196 cm^3^/g), due to the smallest HBA and HBD structures in DES, and the ability to fill both smaller and larger pores. Comparison of results obtained for SiO_2_ impregnated with ChCl: Lev and TPABr: Lev indicates that the type of HBA has a decisive impact on surface area and pore volume. This allows the formulation of the theory that as the aliphatic chain length increases in the HBA structure, the surface area and pore volume decreases. On the other hand, based on a comparison of the results for DES composed of TPABr and various types of HBD, it can be concluded that the presence of Ph in the DES structure also makes it difficult to fill smaller pores. The use of Lev and LA as HBD does not significantly affect the surface area and pore volume.

### 3.2. Adsorption Process

#### 3.2.1. Effect of Different DESs on the Sum of BTEX Adsorption by SiO_2_-DES

In the study, pure SiO_2_ and four SiO_2_ impregnated by DESs were used to investigate the adsorption dynamic behavior. Figure 5 shows the breakthrough curves and saturation adsorption capacity of the sum of BTEX adsorbed on new sorbents. The results indicate that all breakthrough curves can be divided into three phases including effective adsorption. In the first stage, effective BTEX adsorption occurs, in the second stage the breakthrough bed (the adsorbent no longer adsorbed BTEX) occurs and the third stage when the adsorption capacity of the bed in the adsorber is reached. Only a small concentration of BTEX was detected in the first stage and C_OUT_/C_IN_ was negligible, which showed that the adsorption of BTEX by SiO_2_ and SiO_2_ impregnated by DESs could be a fast and effective process. In the adsorption process, the time of breakthrough time was considered to be the point at which concentration of outlet BTEX was 95% of the inlet concentration. The breakthrough times of all sorption materials for BTEX adsorption at the first phase match the following order: SiO_2_-TPABr:LA > SiO_2_-TPABr:Lev > SiO_2_-ChCl:Lev > SiO_2_-TPABr:Ph > SiO_2_. The equilibrium adsorption capacities of SiO_2_-TPABr:LA, SiO_2_-TPABr:Lev, SiO_2_-ChCl:Lev, SiO_2_-TPABr:Ph, and SiO_2_ were 43.1, 147.5, 178.2, 218.8, and 254.9 mg/g, respectively.

The obtained adsorption results indicated that the type of HBA influence on BTEX adsorption capacity. The comparison of BTEX adsorption capacity using pure silica gel and impregnated silica gel by DES composed of levulinic acid, indicate that the BTEX solubility is greater using TPA-Br (218.8 mg/g), than using ChCl (178.2 mg/g). It can be caused by many factors, i.e., quaternary ammonium salts alkyl chain length, type of counterion in HBA structure (Br^−^ or Cl^−^), different charge density on the ammonium, or asymmetry in ChCl ammonium with an −OH in the longest branch. However, more studies are needed to determine which of these factors has the greatest impact on adsorption capacity. Studies on various HBDs show that the main effect on the adsorption capacity has HBD structure, including the occurrence of the benzene ring, and number of −OH, −COOH groups. Depending on the HBD structure, there are various possible interactions between BTEX and new adsorbents. The obtained results indicate that the π–π conjugated bond of the benzene ring in the phenol structure affects the lower BTEX adsorption capacity. This suggests that π–π conjugated bonds do not play a relevant role in BTEX adsorption. While, the increased adsorption capacity of BTEX was observed for SiO_2_-TPABr:LA (254.9 mg/g) and for SiO_2_-TPABr:Lev (218.8 mg/g). Higher adsorption capacity suggests that the additional −OH group in structure HBD increases the affinity of SiO2-TPABr:LA to BTEX, due to the possibility of hydrogen bond formation [50].

#### 3.2.2. Effect of BTEX Initial Concentration

The impact of the initial amount of BTEX on adsorption capacity and adsorption rate was also investigated in a range of 50–300 mg/m^3^ (Figure 6). The results indicate that the breakthrough time decreased from 68 to 52 min, 228 to 179 min, 257 to 216 min, 322 to 265 min, and 366 to 309 min for SiO_2_, SiO_2_-TPABr:Ph, SiO_2_-ChCl:Lev, SiO_2_-TPABr:Lev, and SiO_2_-TPABr:LA, respectively, with the increase in initial BTEX concentration from 50 to 300 mg/m^3^. However, adsorption capacity increased from 9.3 to 43.1 mg/g, 31.4 to 147.5 mg/g, 35.3 to 178.2 mg/g, 44.3 to 218.8 mg/g, and 50.3 to 254.9 mg/g for SiO_2_, SiO_2_-TPABr:Ph, SiO_2_-ChCl:Lev, SiO_2_-TPABr:Lev, and SiO_2_-TPABr:LA, respectively which was attributed to the enhanced driving force to diffusion. A detailed list of SiO_2_ and SiO_2_-DESs adsorption capacity values depending on the initial concentration of BTEX is presented in the Table 3.

In addition to simultaneous co-adsorption of benzene, toluene, ethylbenzene, and p-xylene adsorption of the single components was investigated for all SiO_2_ impregnated by DES and pure SiO_2_. The experimental results revealed that the kind of compounds had only a minor effect on adsorption efficiency. However, the adsorption capacities followed the order of p-xylene > ethylbenzene > toluene> benzene (Table 4). Similar results were obtained for all adsorbents.

#### 3.2.3. Effect of Flow Rate

Gas (containing BTEX) flow rate also have a meaningful influence on adsorption process. (Figure 7). The results indicate that the greater the flow rate of gas, the smaller the breakthrough time. A breakthrough times reduced from 55 to 52 min, 212 to 179 min, 235 to 216 min, 292 to 265 min, and 341 to 309 min for SiO_2_, SiO_2_-TPABr:Ph, SiO_2_-ChCl:Lev, SiO_2_-TPABr:Lev, and SiO_2_-TPABr:LA, respectively, with the increase flow rate from 1 to 5 m^3^/h. On the other hand, adsorption capacity increased from 9.1 to 43.1 mg/g, 35.0 to 147.5 mg/g, 38.8 to 178.2 mg/g, 40.0 to 218.8 mg/g, and 56.3 to 254.9 mg/g for SiO_2_, SiO_2_-TPABr:Ph, SiO_2_-ChCl:Lev, SiO_2_-TPABr:Lev, and SiO_2_-TPABr:LA. The result indicated that the increasing flow rate of enhanced the BTEX adsorption capacity. A detailed list of SiO_2_ and SiO_2_-DESs adsorption capacity values depending on the flow rate is presented in the Table 5.

#### 3.2.4. Recycling and Regeneration of SiO_2_-DES

From an economic point of view, the possibility of recycling and regeneration of the adsorbents is one of the most important parameters. In the studies, the adsorption processes were done after complete desorption of BTEX, which was carried using an inert gas purge flow (5 L/h), at 80 °C, to flush the BTEX off the adsorbent material for 120 min. From the adsorption breakthrough curves, the breakthrough times after the fifth cycle (adsorption/desorption) shifted a maximum of 3 min relative to values obtained for fresh adsorbents. This suggests nearly no loss of SiO_2_ and SiO_2_-DES adsorption capacity after five repeated adsorption/desorption cycles (Table 6 and Figure 8). The obtained results indicate that all studied SiO_2_ and SiO_2_-DES adsorbents maintain its stability and could be re-used for many cycles.

## 4. Conclusions

In the paper, four DES including TPABr:Lev, TPABr:LA, TPABr, and ChCl:Lev were combined with the porous silica gel to prepare SiO_2_ supported deep eutectic solvents, by incipient impregnation method. Structures and properties of new adsorbents were characterized by FTIR-ATR, TG/DTG, SEM, XRD, and BET. Compared with pure SiO_2_, the adsorption capacities of SiO_2_ impregnated by DES are all increased, indicating that the deep eutectic solvents treatment significantly increased the adsorption capacity of silica gel. This result can be explained by specific interactions between DESs and BTEX, i.e., hydrogen bonds interaction. However, further analysis using, e.g., molecular modeling is necessary to explain all potential interactions affecting the BTEX adsorption process using SiO_2_-DES.

In addition, the results indicated SiO_2_-TPABr:LA adsorbent as a potential candidate for BTEX capture alternative as its adsorption capacities were higher than the pure SiO_2_ and other developed SiO_2_-DES adsorbents. The ability and effectiveness of this adsorbent in capturing BTEX at room temperature and atmospheric pressure makes it a practical material for industrial processes. In addition, all adsorbents effectively retain its stability and could be re-used for many cycles.

## Figures and Tables

**Figure 1 materials-13-01894-f001:**
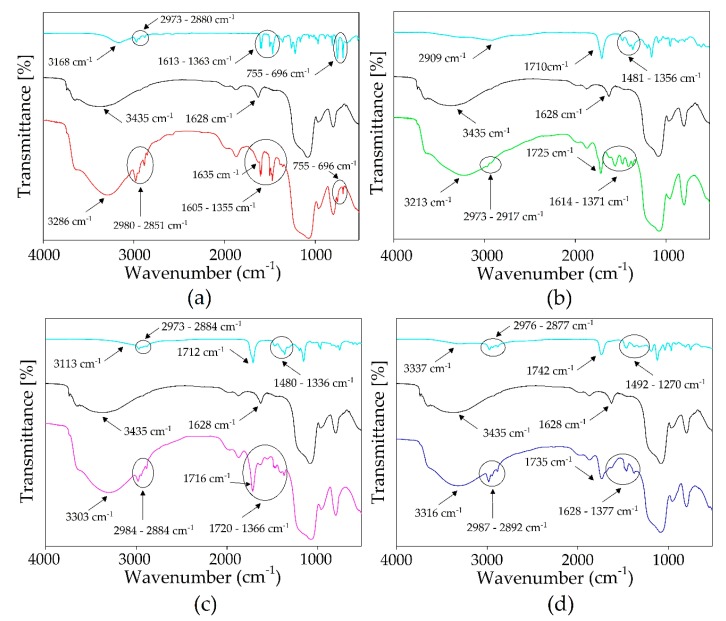
The FT-IR-ATR spectra of (**a**) SiO_2_-TPABr:Ph; (**b**) SiO_2_-ChCl:Lev; (**c**) SiO_2_-TPABr:Lev; (**d**) SiO_2_-TPABr:LA. The first spectrum corresponds to DES, the middle SiO_2_, the last to SiO_2_-DES.

**Figure 2 materials-13-01894-f002:**
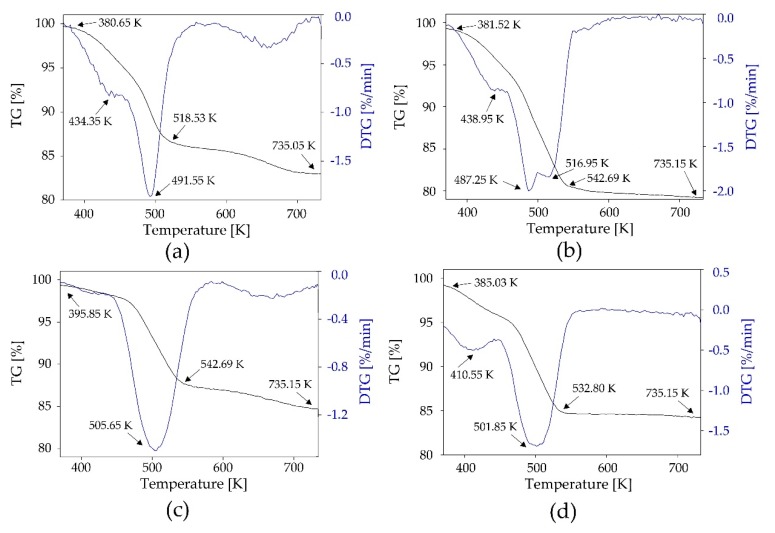
TG/DTG thermograms for (**a**) SiO2-ChCl:Lev; (**b**) SiO2-TPABr:Lev; (**c**) SiO2-TPABr:LA; (**d**) SiO2-TPABr:Ph.

**Figure 3 materials-13-01894-f003:**
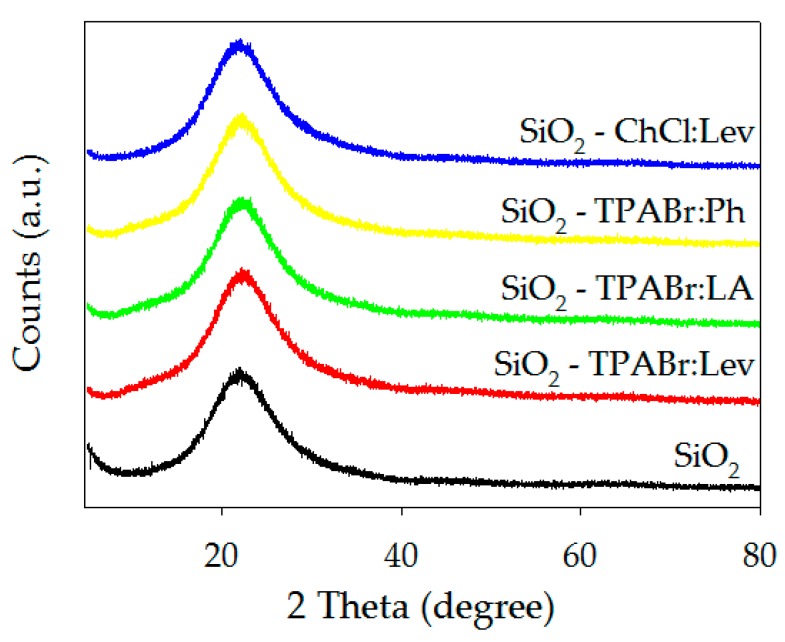
XRD analysis of SiO_2_ and SiO_2_ impregnated with DESs.

**Figure 4 materials-13-01894-f004:**
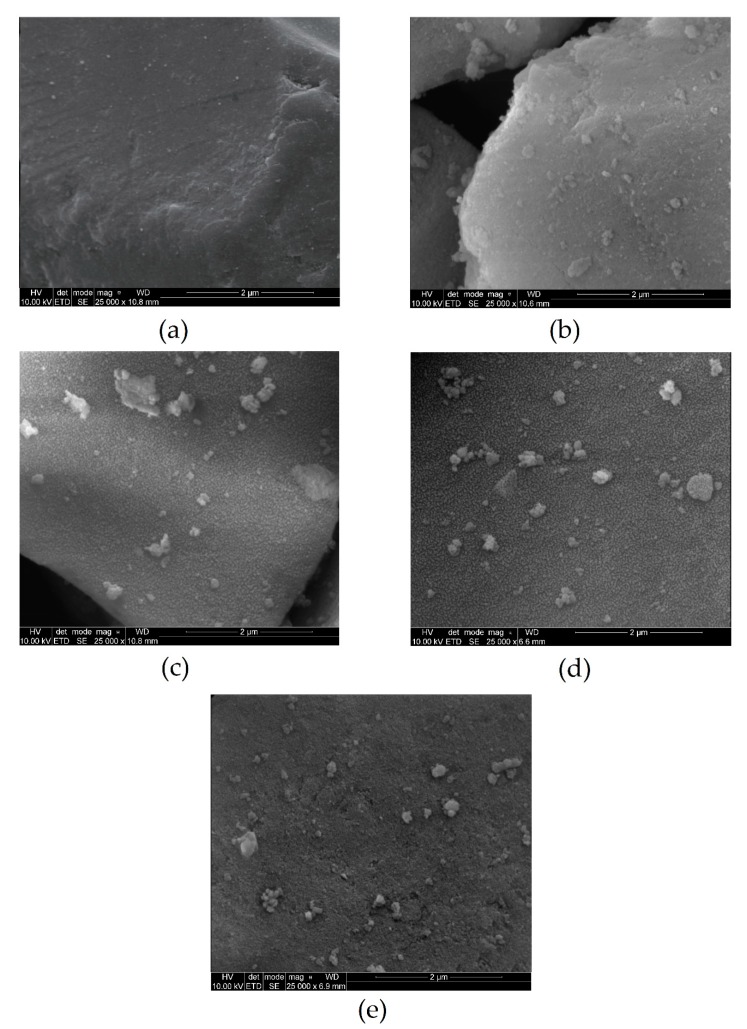
SEM images of investigated adsorbents: (**a**) Silica gel (SiO_2_); (**b**) SiO_2_-ChCl:Lev; (**c**) SiO_2_-TPABr:LA; (**d**) SiO_2_-TPABr:Lev; (**e**) SiO_2_-TPABr:Ph.

**Figure 5 materials-13-01894-f005:**
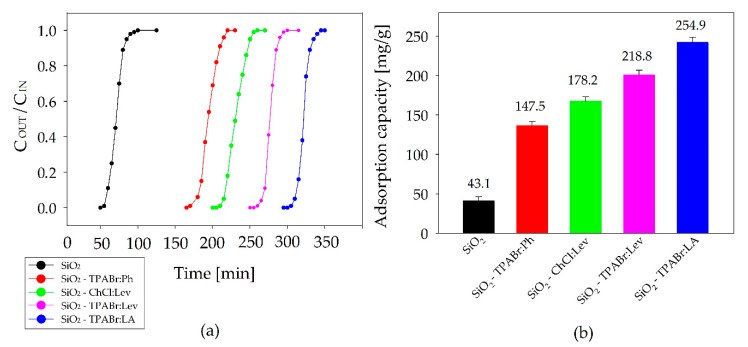
(**a**) Adsorption breakthrough curves of different types of adsorbents; (**b**) Adsorption capacity (C_IN_ = 300 mg/m^3^; F = 5 m^3^/h; m = 30 g; T = 20 °C; p = 1 atm).

**Figure 6 materials-13-01894-f006:**
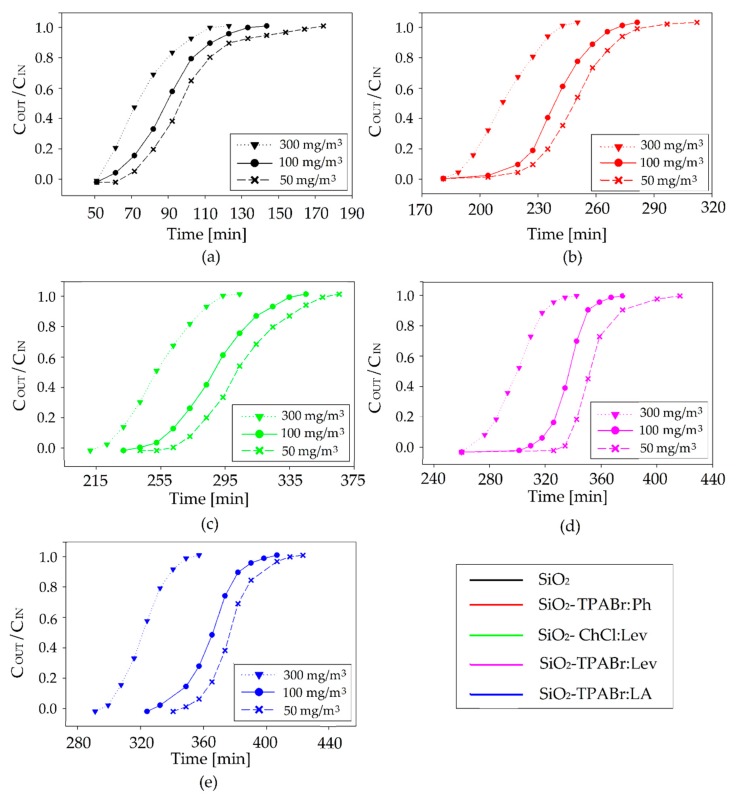
Effect of BTEX initial concentration on adsorption breakthrough curves of (**a**) SiO_2_, (**b**) SiO_2_-TPABr:Ph, (**c**) SiO_2_-ChCl:Lev, (**d**) SiO_2_-TPABr:Lev, (**e**) SiO_2_-TPABr:LA (F = 5 m^3^/h; m = 30 g; T = 20 °C; p = 1 atm).

**Figure 7 materials-13-01894-f007:**
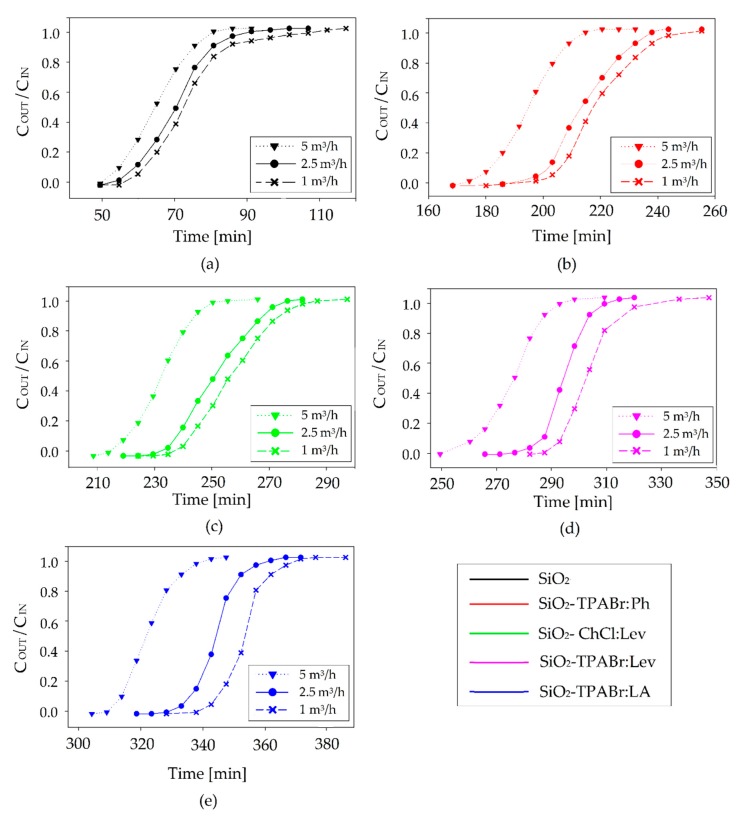
Effect of flow rate on adsorption breakthrough curves of (**a**) SiO_2_, (**b**) SiO_2_-TPABr:Ph, (**c**) SiO_2_-ChCl:Lev, (**d**) SiO_2_-TPABr:Lev, (**e**) SiO_2_-TPABr:LA (C_IN_ = 300 mg/m^3^; m = 30 g; T = 20 °C; p = 1 atm).

**Figure 8 materials-13-01894-f008:**
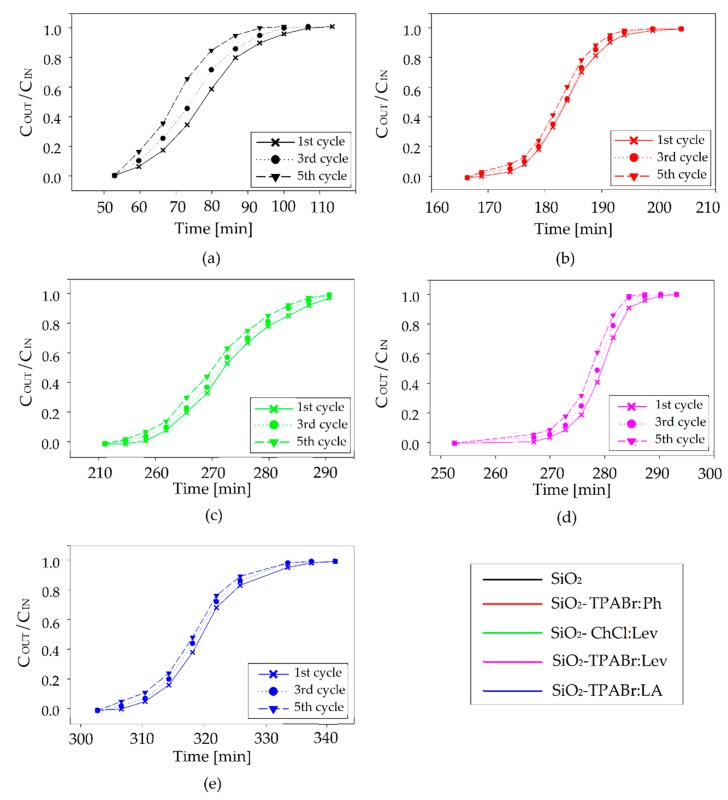
The effect of regeneration cycles on the on adsorption breakthrough curves of (**a**) SiO_2_, (**b**) SiO_2_-TPABr:Ph, (**c**) SiO_2_-ChCl:Lev, (**d**) SiO_2_-TPABr:Lev, (**e**) SiO_2_-TPABr:LA (C_IN_ = 300 mg/m^3^; F = 5 m^3^/h; m = 30 g; T = 20 °C; p = 1 atm).

**Table 1 materials-13-01894-t001:** Characteristics of deep eutectic solvents (DES) loaded silica gel.

No.	Type of Adsorbent	Type of DES (HBA:HBD mol ratio)	Melting Points of DES (°C)	Viscosity of DES at 25 °C (mPas)	Abbreviation of New Adsorbent	Impregnation Solution DES/MeOH (wt.% DES)	m_DES_ (g)
1	SiO_2_	−	−	−	SiO_2_	85%	−
2	SiO_2_	ChCl:Lev (1:2)	Liquid at RT ^1)^	255.8 ^1^	SiO_2_-ChCl:Lev	85%	0.58
3	SiO_2_	TPABr:LA (1:2)	−11	>600	SiO_2_-TPABr:LA	85%	0.57
4	SiO_2_	TPABr:Lev (1:3)	−7	>600	SiO_2_-TPABr:Lev	85%	0.61
5	SiO_2_	TPABr:Ph (1:2)	24	>600	SiO_2_-TPABr:Ph	85%	0.60

^1.^ [43,44]; RT—room temperature.

**Table 2 materials-13-01894-t002:** Brunauer–Emmet–Teller (BET) surface area and pore volume of different types of adsorbents.

No.	Adsorbent	BET Surface Area (m^2^/g)	Pore Volume (cm^3^/g)
1	SiO_2_	299.41	0.1484
2	SiO_2_-ChCl:Lev	248.11	0.1196
3	SiO_2_-TPABr:LA	257.71	0.1238
4	SiO_2_-TPABr:Lev	253.45	0.1228
5	SiO_2_-TPABr:Ph	281.29	0.1358

**Table 3 materials-13-01894-t003:** List of SiO_2_ and SiO_2_-DESs adsorption capacity values depending on the initial concentration of BTEX.

Adsorbent	Initial Concentration (mg/m^3^)		
	50	100	300
SiO_2_	9.3	26.1	43.1
SiO_2_-TPABr:Ph	31.4	91.8	147.5
SiO_2_-ChCl:Lev	35.3	100.1	178.2
SiO_2_-TPABr:Lev	44.3	151.4	218.8
SiO_2_-TPABr:LA	50.3	188.5	254.9

**Table 4 materials-13-01894-t004:** List of SiO_2_ and SiO_2_-DESs adsorption capacity values depending on the kind of BTEX structure.

Adsorbent	Initial Concentration (mg/m^3^)			
	Benzene	Toluene	Ethylbenzene	p-xylene
SiO_2_	40.1	42.4	44.7	46.2
SiO_2_-TPABr:Ph	138.2	140.8	150.1	154.2
SiO_2_-ChCl:Lev	155.9	165.4	174.9	189.9
SiO_2_-TPABr:Lev	202.7	211.2	215.5	228.4
SiO_2_-TPABr:LA	222.3	234.1	259.7	271.3

**Table 5 materials-13-01894-t005:** List of SiO_2_ and SiO_2_-DESs adsorption capacity values depending on flow rate.

Adsorbent	Flow Rate (m^3^/h)		
	1	2.5	5
SiO_2_	9.1	36.2	43.1
SiO_2_-TPABr:Ph	35.0	114.1	147.5
SiO_2_-ChCl:Lev	38.8	127.4	178.2
SiO_2_-TPABr:Lev	40.0	181.6	218.8
SiO_2_-TPABr:LA	56.3	200.1	254.9

**Table 6 materials-13-01894-t006:** List of SiO_2_ and SiO_2_-DESs adsorption capacity values depending on the number of cycles.

Adsorbent	Number of Cycles		
	1st	3rd	5th
SiO_2_	43.1	41.2	39.8
SiO_2_-TPABr:Ph	147.5	146.0	145.4
SiO_2_-ChCl:Lev	178.2	177.2	176.4
SiO_2_-TPABr:Lev	218.8	216.1	215.5
SiO_2_-TPABr:LA	254.9	252.7	251.4

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
