# Peer review of "Silica Gel Impregnated by Deep Eutectic Solvents for Adsorptive Removal of BTEX from Gas Streams"

_materials, 2020, doi:10.3390/ma13081894_

Round 1

Reviewer 1 Report

A strong revision is needed as there are many problems in the paper that must be considered prior of considering it for publication. The paper is well written and the experiments are well conducted, but the conclusions are bad considered from this set of experiments.

  1. The authors wrote in the abstract that TPABr:LA absorbed DES is the best in their set and that “This result can be explained by the combination of surface chemistry, pore volume and pore structure of the impregnated SiO2 material, and other specific interaction between DESs and BTEX i.e. hydrogen bonds and van der Waals interaction.” But this is not true as it emerged from their measures as the properties of this SiO2-DES are in the middle between all the others. In all the paper there is a lack of any structure/activity good correlation between the observed structural results and the adsorption capacities. It seems there are other factors that can determine this capacity but the authors did not measure them. Aromatic compounds are showing a great affinity for this kind of liquids as they can be extracted/removed from a great number of different matrixes (some literature of these experiments must be reported in the introduction). However, nothing has been reported yet about the mechanism of these extractions, as it seems a difficult matter to be solved (can it be considered that aromatic rings are HBA therefore they can participate to the H-bonds network?). In this paper it seems there’s again the same matter. The authors have to change all the S/A considerations as they are superficially approached and there are many wrong conclusions.

  1. I do believe the term “IONIC Deep Eutectic Solvents” used by the authors in many parts of the manuscript is wrong. Even if a DES can be formed by ionic species (such as ChCl or TBABr as the ones used by the authors) this term is incorrect since a DES can be formed by non-ionic species, by zwitterionic species, or by amphoteric ones as well. I believe is a too specific term, “Deep Eutectic Solvent” is enough. Moreover, the term “ionic” in DESs’ literature could be related to Walden plot measures and this could generate confusion.

  1. The authors used four DESs for their studies: the ones already reported in literature must be cited in references, the ones not reported (if there are) must be characterized at least with an eutectic profile.

  1. I think the part about the mixtures made, their loading on silica, the abbreviations (table 1 and few words about their preparations) can be moved as first paragraph of results and discussion section, leaving the experimental details only in experimental.

  1. In equation 1 the “2” in subscript in SiO2 must be checked.

  1. In line 164 the authors wrote “oven temperature program was 110 °C” what does it mean for a GC analysis? Isn’t it better to report the initial and final T, increase of T/min and times for initial and final stages as usually reported for a GC analysis?

  1. In line 181 the authors report about C=O stretching peaks in FT-IR on phenol-based systems, what does it mean?

  1. In line 253 the authors wrote “This allows the formulation of the theory that as the aliphatic chain length increases in the HBA structure, the surface area and pore volume decreases.” This seems a little bit too much: a theory based on a single data on the alkyl chains of 3 methylene groups.

  1. Phenol-based DESs showed the lower Adsorption Capacity in the set, this indicates that the pie-pie interactions could not play a relevant role in aromatic absorption, but H-bonds are preferred, this must be underlined in the structure/activity considerations the authors could write.

  1. In line 295 the authors wrote “Similar results were obtained for all adsorbents.” I think it is better to report a table or some data about it (also in supporting) instead of this too simple sentence.

  1. The graphs in Figures 6,7 and 8 can be better rewritten keeping the different kind of lines but with differently-shaped indicators because it is a little bit hard to detect the differences in the samples.

Author Response

Dear Reviewer,

Thank you for your consideration and evaluation of the paper. All corrections raised by the Reviewer were thoroughly considered. Detailed explanations can be found below.

  • The authors wrote in the abstract that TPABr:LA absorbed DES is the best in their set and that “This result can be explained by the combination of surface chemistry, pore volume and pore structure of the impregnated SiO2 material, and other specific interaction between DESs and BTEX i.e. hydrogen bonds and van der Waals interaction.” But this is not true as it emerged from their measures as the properties of this SiO2-DES are in the middle between all the others. In all the paper there is a lack of any structure/activity good correlation between the observed structural results and the adsorption capacities. It seems there are other factors that can determine this capacity but the authors did not measure them. Aromatic compounds are showing a great affinity for this kind of liquids as they can be extracted/removed from a great number of different matrixes (some literature of these experiments must be reported in the introduction). However, nothing has been reported yet about the mechanism of these extractions, as it seems a difficult matter to be solved (can it be considered that aromatic rings are HBA therefore they can participate to the H-bonds network?). In this paper it seems there’s again the same matter. The authors have to change all the S/A considerations as they are superficially approached and there are many wrong conclusions.

We have carefully reviewed the manuscript, corrected it and added appropriate explanations to many points of the manuscript. We tried to include all valuable comments. Due to the very complex comments in the first point, we have included all the explanations in the manuscript.

  • I do believe the term “IONIC Deep Eutectic Solvents” used by the authors in many parts of the manuscript is wrong. Even if a DES can be formed by ionic species (such as ChCl or TBABr as the ones used by the authors) this term is incorrect since a DES can be formed by non-ionic species, by zwitterionic species, or by amphoteric ones as well. I believe is a too specific term, “Deep Eutectic Solvent” is enough. Moreover, the term “ionic” in DESs’ literature could be related to Walden plot measures and this could generate confusion.

The term “IONIC Deep Eutectic Solvents”  was corrected in the article according to the Reviewer's comment.

  • The authors used four DESs for their studies: the ones already reported in literature must be cited in references, the ones not reported (if there are) must be characterized at least with an eutectic profile.

The information about DES was corrected and supplemented in Table 1 according to the Reviewer's comment.

  • I think the part about the mixtures made, their loading on silica, the abbreviations (table 1 and few words about their preparations) can be moved as first paragraph of results and discussion section, leaving the experimental details only in experimental.

The characteristics of DES have been moved to the first point "Results and discussion" according to the Reviewer's comment.

“In the studies, deep eutectic solvents occurring already in the literature were synthesized. The silica gel used in this studies is characterized by a high surface area and high thermal stability that can be used in various applications [41,42]. A silica gel with high porosity works as a supporter of DES in the impregnation process. This is because of the fact in the silica gel can bond with DES physically through the formation of a hydrogen bond or van der Waals interaction [43]. The amount of DES (mDES) loaded on the silica gel was determined using equation (1). The list and characteristics of new sorbents are presented in Table 1. Liquid absorption (DES) material integrated with porous silica gel is expected to increase BTEX adsorption capacity. Additionally, impregnation on solid materials appears as a good platform, in order to overcome limitations related to DES i.e. a high viscosity.”

Table 1 Characteristics of deep eutectic solvents (DES) loaded silica gel.

No.

Type of adsorbent

Type of DES (HBA:HBD mol ratio)

Melting points (°C)

Abbreviation of new adsorbent

Impregnation solution

DES/MeOH
(wt% DES)

m DES

[g]

1

SiO2

-

-

SiO2

85 %

-

2

SiO2

ChCl:Lev (1:2)

Liquid at RT1)

SiO2 - ChCl:Lev

85 %

0.58

3

SiO2

TPABr:LA (1:2)

-11

SiO2 - TPABr:LA

85 %

0.57

4

SiO2

TPABr:Lev (1:3)

-7

SiO2 - TPABr:Lev

85 %

0.61

5

SiO2

TPABr:Ph (1:2)

24

SiO2 - TPABr:Ph

85 %

0.60

1) RT – room temperature [44,45]

  • In equation 1 the “2” in subscript in SiO2 must be checked.

The subscript was corrected according to the Reviewer comment.

  • In line 164 the authors wrote “oven temperature program was 110 °C” what does it mean for a GC analysis? Isn’t it better to report the initial and final T, increase of T/min and times for initial and final stages as usually reported for a GC analysis?

In the studies, we used isotherm program (11O°C). This information was added to the manuscript. 

 “The isotherm temperature oven in the gas chromatograph was 110 °C”.

  • In line 181 the authors report about C=O stretching peaks in FT-IR on phenol-based systems, what does it mean?

The sentence was corrected according to the Reviewer comment.

“The characteristic shifts of the carbon-carbon double bonds (C = C) towards the higher wavenumber can be observed in the Figure 1a around 1635 cm-1, while the shifts of the carbonyl group (C = O) towards the higher wavenumber can be observed around 1725 cm-1 (Figure 1b), 1716 cm-1 (Figure 1c), 1735 cm-1 (Figure 1d) and the extra peaks in the range of 1605-1355 cm-1(Figure 1A), 1614-1371 cm-1 (Figure 1b), 1720-1366 cm-1 (Figure 1c) and 1628-1377 cm-1 (Figure 1d), that was attributed to asymmetric and symmetric CH2 vibrations, C-O stretching bond, and C-H bonding vibration in the alkaline chain.”

  • In line 253 the authors wrote “This allows the formulation of the theory that as the aliphatic chain length increases in the HBA structure, the surface area and pore volume decreases.” This seems a little bit too much: a theory based on a single data on the alkyl chains of 3 methylene groups.

The sentence was corrected according to the Reviewer comment.

“This allows the formulation of the application that as the aliphatic chain length increases in the HBA structure, the surface area and pore volume decreases”.

  • Phenol-based DESs showed the lower Adsorption Capacity in the set, this indicates that the pie-pie interactions could not play a relevant role in aromatic absorption, but H-bonds are preferred, this must be underlined in the structure/activity considerations the authors could write.

The manuscript was appended according to the Reviewer comment. 

“The obtained adsorption results indicated that the length of the HBA influence BTEX adsorption capacity. The comparison of BTEX adsorption capacity using pure silica gel and impregnated silica gel by DES composed of levulinic acid indicates that the BTEX solubility increases with increasing alkyl chain length of the HBA. The BTEX adsorption capacity increased from 178.2 to 218.8 mg/g when the alkyl chain length increased from methyl to propyl (i.e. from choline to tetrapropylammonium). By increasing the alkyl chain length of HBA, the increase of the molar volume and the free volume increase, which results in higher BTEX solubility. Studies on the use of various HBDs show that the main effect on the adsorption capacity has HBD structure (ring, number of OH groups). Depending on the HBD structure, there are various possible interactions between the new adsorbents and BTEX. The obtained results indicate that the π– π conjugated bond of the benzene ring in the phenol structure has shown the lower BTEX adsorption capacity. This suggests that π– π conjugated bonds do not play a relevant role in BTEX adsorption. The increased adsorption capacity BTEX has observed using silica gel impregnated by TPABr:LA, which was 254.9 mg/g and while for SiO2 impregnated by TPABr:Lev was 218.8 mg/g. Increased sorption capacity suggests that the additional OH group in structure HBD increases the affinity SiO2-TPABr:LA to BTEX, due to possibility of OH-π bonds formation [45]. In addition, the additional OH group provide to the formation of strong hydrogen bonding and electrostatic interactions (i.e. interaction van der Waals), which probably played the main influences on the adsorption process efficiency.”

  • In line 295 the authors wrote “Similar results were obtained for all adsorbents.” I think it is better to report a table or some data about it (also in supporting) instead of this too simple sentence.

The authors have added tables with adsorption capacity values to all optimized conditions to ensure readability.

  • The graphs in Figures 6,7 and 8 can be better rewritten keeping the different kind of lines but with differently-shaped indicators because it is a little bit hard to detect the differences in the samples.

The descriptions in Figures 6,7 and 8 were corrected according to the Reviewer comment. 

Reviewer 2 Report

The paper synthesized new adsorbents based on silica gel impregnated with DESs to increase BTEX adsorption efficiency from gas streams. This study is clear, well-organized and of great interests in the adsorption filed. I recommend publication after addressing the following minor issue.

  1. The order of BTEX in the abstract should be rearranged.

  1. A few technologies are mentioned in the introduction. The authors stated that one of the most popular is adsorption. What is the weakness of other technologies? A comparison will make the statement more convincing.

  1. In several works have described the impregnation of activated carbon with the use of choline chloride : urea (1:2 molar ratio), silica gel with choline chloride:glycerol (1:2) [28], and choline chloride:urea (1:2 molar ratio) [29] and using them to remove carbon dioxide from the gas streams. This sentence should have a subject.

  1. There are also some careless writings such as “The mixture was agitated 2 hours at 25 °C”, “In order to XRD analysis”.

  1. For the nitrogen adsorption test, was degassing with nitrogen at 200 °C for 2 hours sufficient to remove all the pre-sorbed matters?

  1. 77 K is not equal to -195 °C.

  1. In TGA test, if there are evaporations of water and methanol, I suggest showing the results from 100 degree and treat that mass as the original mass.

  1. Did authors turn on EDX when conducting SEM? If so, it would be interesting to tell what the islands are on the surface in Fig. 4b-e.

  1. 5b is called saturation capacity. I understand it was calculated from Eq.2. When doing the integration, what is the upper limit of time t? For example of SiO2, was it 50min or 90 min? If it is 90min, it is ok to call it saturation capacity. But readers are more interested in the breakthrough capacity. That is the integration up to 50th min.

  1. In Fig. 6, it would be better to show an integrated capacity like what was shown in Fig. 5. The same suggestion for Fig. 7 and 8.

Author Response

Dear Reviewer,

Thank you for your consideration and evaluation of the paper. All corrections raised by the Reviewer were thoroughly considered. Detailed explanations can be found below.

  • The order of BTEX in the abstract should be rearranged.

 The order of BTEX in the abstract was corrected according to Reviewer comment.

  • A few technologies are mentioned in the introduction. The authors stated that one of the most popular is adsorption. What is the weakness of other technologies? A comparison will make the statement more convincing.

The weakness of other technologies were added to the manuscript, according to Reviewer comment.

“Chemical and physical absorption have several disadvantages i.e. foaming possibility, inefficient for numerous groups of chemical compounds due to their limited solubility, and high amount of chemical solvents (sometimes toxic and corrosive compounds) are requiring. Pressure swing adsorption and cryogenic separation ensures high efficiency of removing contaminants from gas streams. However, high electricity consumption, expensive investment and operation often prevent practical use. In addition, pre-treatment is required. Membrane separation is expensive process and not suitable for high polluted gasses. In turn, biological separation characterized by low impurities removal efficiency and longtime operation process.”

  • In several works have described the impregnation of activated carbon with the use of choline chloride : urea (1:2 molar ratio), silica gel with choline chloride:glycerol (1:2) [28], and choline chloride:urea (1:2 molar ratio) [29] and using them to remove carbon dioxide from the gas streams. This sentence should have a subject.

The sentence was corrected according to Reviewer comment.

  • There are also some careless writings such as “The mixture was agitated 2 hours at 25 °C”, “In order to XRD analysis”.

 The sentences were corrected according to Reviewer comment.

“The adsorbents were agitated 2 hours at 25 °C and dried in a vacuum oven at 90 °C for 6 hours”.

“Samples of pure and modified silica gel were tested in the range of 5 to 80° in steps of 0.01° using XRD instrument”.

  • For the nitrogen adsorption test, was degassing with nitrogen at 200 °C for 2 hours sufficient to remove all the pre-sorbed matters?

All samples were pretreated by degassing with nitrogen at 200° C for 2 hours to remove adsorbed contaminants acquired, which typically is just water and carbon dioxide from the atmospheric exposure. For some samples, sensitive to high temperatures (e.g., pharmaceuticals, organic materials), the temperature would have to be much lower, but it is appropriate for the DES samples.

To perform the best quality experiments and verify whether the dedicated time and temperature are sufficient to remove all pre-sorbed matter authors performed additional experiments on selected DES samples to verify whether the mass of the sample would change after treatment for a longer time than 2hours. The results indicated that the mass of the samples has not changed, so we assumed the applied conditions were sufficient to proceed with future experimental work. 

Moreover, before performing any experimental work, we analyzed some previously documented papers concerning BET experiments. Around many, authors applied similar conditions to analyze the surface area and pore volume of substances such as silica, alumina/silica and boehmite/silica (Linsha et al., 2013), aluminosilicates (Sequaris et al., 2012), sol-gel silica (Nampi et al., 2013),  KTaO3, CdS, MoS2, Semiconductors  (Bajorowicz et al., 2014), TiO2 – photocatalyst (Zielińska-Jurek et al., 2017), and other types of catalysts (Wysocka 2019).

  1. Wysocka, I., Hupka, J., Rogala, A., 2019, Catalytic Activity of Nickel and Ruthenium–Nickel Catalysts Supported on SiO2, ZrO2, Al2O3, and MgAl2O4 in a Dry Reforming Process, Catalysts 2019, 9, 540, doi:10.3390/catal9060540
  2. Zielińska-Jurek, A., Bielan, Z., Dudziak, S., Wolak, I., Sobczak, Z., Klimczuk, T., Nowaczyk, G., Hupka, J., 2017, Design and Application of Magnetic Photocatalysts for Water Treatment. The Effect of Particle Charge on Surface Functionality, Catalyst, doi:10.3390/catal7120360
  3. Séquaris, J-M, Klumpp, E., Vereecken, H., 2013, Colloidal properties and potential release of water-dispersible colloids in an agricultural soil depth profile, Geoderma, 193–194. 
  4. Linsha, V., Suchithra, Peer M.,A., Ananthakumar, S., 2013, Amine-grafted alumino-siloxane hybrid porous granular media: A potential sol–gel sorbent for treating hazardous Cr(VI) in aqueous environment, Chemical Engineering Journal (220) 244–253.
  5. Nampi,P.,P., Kartha, C.,C., Jose, G., Kumar A., P.,R.,  Anilkumar, T., Varma, H., 2013,  Sol–gel nanoporous silica as substrate for immobilization of conjugated biomolecules for application as fluorescence resonance energy transfer (FRET) based biosensor, Sensors and Actuators B: Chemical, 185.
  6. Bajorowicz B., Cybula A.,  Winiarski M., J., Klimczuk T., Zaleska, A., 2014, Surface Properties and Photocatalytic Activity of KTaO3, CdS, MoS2 Semiconductors and Their Binary and Ternary Semiconductor Composites Molecules 19(9), 15339-15360; doi.org/10.3390/molecules190915339.

  • 77 K is not equal to -195 °C.

The value of temperature was corrected, according to Reviewer comment. The temperature of -195°C  was changed to -196.15 °C.

  • In TGA test, if there are evaporations of water and methanol, I suggest showing the results from 100 degree and treat that mass as the original mass.

 All TGA/DTG thermograms were corrected according to reviewer comments.

  • Did authors turn on EDX when conducting SEM? If so, it would be interesting to tell what the islands are on the surface in Fig. 4b-e.

Unfortunately, we did not conduct such research. However, we agree with the Reviewer that this studies will be very interesting and valuable, so we will consider this suggestion in future studies.

  • 5b is called saturation capacity. I understand it was calculated from Eq.2. When doing the integration, what is the upper limit of time t? For example of SiO2, was it 50min or 90 min? If it is 90min, it is ok to call it saturation capacity. But readers are more interested in the breakthrough capacity. That is the integration up to 50th min.

Adsorption capacity was calculated using Eq. 2. In the adsorption process, for the calculation the time of breakthrough was considered to be the point at which concentration of outlet BTEX was 95% of the inlet concentration. For example, for SiO2 it was approximately 50 min.

  • In Fig. 6, it would be better to show an integrated capacity like what was shown in Fig. 5. The same suggestion for Fig. 7 and 8.

The authors have added tables with adsorption capacity values to all optimized conditions to ensure readability.

Round 2

Reviewer 1 Report

The authors correctly answered to all the comments made in the previous revision, however there are still some problems:

- On the previous version of the paper it was pointed out that the experimental set was not proper to determine the structure/activity relationships as the results did not correlate with any structural feature of the systems. In many parts of the paper (for example in the conclusion section) the authors properly described the results and didn’t write anything that could be bad interpreted, however in lines 482-488 there are still some bad-interpreted results: how can ChCl and TPABr HBAs can be compared in terms of their length? The two molecules are different in many parts and they cannot be compared considering only their length as they have different counterion, different charge density on the ammonium, there’s an asymmetry in ChCl ammonium with an -OH in the longest branch etc. etc. I do believe it’s better to reconsider this part in the corrected paper.  

- In line 508: van der Waals forces are different from hydrogen bonding and electrostatic interactions.

- Lines 331-339 and lines 482-509 have a bad English language, there are many mistakes.

- “OH-π interaction” seems like alcohol-π interaction and it’s not correct, it’s better to write it differently as it is a hydrogen bond. In many other parts of the paper the hydrogen bonds in the DESs were written as “OH” parts involved, it’s not correct since the HBDs are carboxylic acids.

Author Response

Dear Reviewer,

Thank you for your consideration and re-evaluation of the paper. All corrections raised by the Reviewer were thoroughly considered. Detailed explanations can be found below.

  • On the previous version of the paper it was pointed out that the experimental set was not proper to determine the structure/activity relationships as the results did not correlate with any structural feature of the systems. In many parts of the paper (for example in the conclusion section) the authors properly described the results and didn’t write anything that could be bad interpreted, however in lines 482-488 there are still some bad-interpreted results: how can ChCl and TPABr HBAs can be compared in terms of their length? The two molecules are different in many parts and they cannot be compared considering only their length as they have different counterion, different charge density on the ammonium, there’s an asymmetry in ChCl ammonium with an -OH in the longest branch etc. etc. I do believe it’s better to reconsider this part in the corrected paper.

This part of the manuscript was corrected according to the Reviewer's comment.

“The obtained adsorption results indicated that the type of HBA influence on BTEX adsorption capacity. The comparison of BTEX adsorption capacity using pure silica gel and impregnated silica gel by DES composed of levulinic acid, indicate that the BTEX solubility is greater using TPA-Br (218.8 mg/g), than using ChCl (178.2 mg/g). It can be caused by many factors i.e. quaternary ammonium salts alkyl chain length, type of counterion in HBA structure (Br- or Cl-), different charge density on the ammonium, or asymmetry in ChCl ammonium with an -OH in the longest branch. However, more studies are needed to determine which of these factors has the greatest impact on adsorption capacity.”

  • In line 508: van der Waals forces are different from hydrogen bonding and electrostatic interactions.

The manuscript was corrected according to the Reviewer's comment.

  • Lines 331-339 and lines 482-509 have a bad English language, there are many mistakes.

The manuscript was carefully checked and corrected according to the Reviewer's comment.

Lines 331-339:

“The silica gel used in these studies is characterized by a high surface area and high thermal stability [41,42]. The high porosity of SiO2 supports the impregnation process due to the fact that the silica gel can bond with DES physically through the formation of a hydrogen bonds or van der Waals interactions [43]. The DESs are characterized by high viscosity (>250 mPas), and a melting point below 25°C. The high viscosity of DESs is unfavorable in many applications i.e. extraction, absorption. While in the process of impregnation, high viscosity enables permanent deposition of DES on the SiO2 surface. The list and characteristics of new adsorbents is presented in Table 1.”

Lines 482-509:

“The obtained adsorption results indicated that the type of HBA influence on BTEX adsorption capacity. The comparison of BTEX adsorption capacity using pure silica gel and impregnated silica gel by DES composed of levulinic acid, indicate that the BTEX solubility is greater using TPA-Br (218.8 mg/g), than using ChCl (178.2 mg/g). It can be caused by many factors i.e. quaternary ammonium salts alkyl chain length, type of counterion in HBA structure (Br- or Cl-), different charge density on the ammonium, or asymmetry in ChCl ammonium with an -OH in the longest branch. However, more studies are needed to determine which of these factors has the greatest impact on adsorption capacity. Studies on various HBDs show that the main effect on the adsorption capacity has HBD structure, including the occurrence of the benzene ring, and number of -OH, -COOH groups. Depending on the HBD structure, there are various possible interactions between BTEX and new adsorbents. The obtained results indicate that the π– π conjugated bond of the benzene ring in the phenol structure affects the lower BTEX adsorption capacity. This suggests that π– π conjugated bonds do not play a relevant role in BTEX adsorption. While, the increased adsorption capacity of BTEX was observed for SiO2-TPABr:LA (254.9 mg/g) and for SiO2-TPABr:Lev (218.8 mg/g). Higher adsorption capacity suggests that the additional -OH group in structure HBD increases the affinity of SiO2-TPABr:LA to BTEX, due to the possibility of hydrogen bond formation [51].”

  • “OH-π interaction” seems like alcohol-π interaction and it’s not correct, it’s better to write it differently as it is a hydrogen bond. In many other parts of the paper the hydrogen bonds in the DESs were written as “OH” parts involved, it’s not correct since the HBDs are carboxylic acids.

The manuscript was carefully checked and corrected according to the Reviewer's comment.